# Association between High HbA1c Levels and Mast Cell Phenotype in the Infrapatellar Fat Pad of Patients with Knee Osteoarthritis

**DOI:** 10.3390/ijms25020877

**Published:** 2024-01-10

**Authors:** Ayumi Tsukada, Ken Takata, Jun Aikawa, Dai Iwase, Manabu Mukai, Yui Uekusa, Yukie Metoki, Gen Inoue, Masayuki Miyagi, Masashi Takaso, Kentaro Uchida

**Affiliations:** 1Department of Orthopedic Surgery, Kitasato University School of Medicine, 1-15-1 Minami-ku Kitasato, Sagamihara City 252-0374, Kanagawa, Japan; amidesutarere9010@yahoo.co.jp (A.T.); kentakata41@yahoo.co.jp (K.T.); jun43814@gmail.com (J.A.); daiiwase19760601@yahoo.co.jp (D.I.); m.manabu0829@hotmail.co.jp (M.M.); uekusa18y@gmail.com (Y.U.); yukiemetoki0826@gmail.com (Y.M.); ginoue@kitasato-u.ac.jp (G.I.); masayuki008@aol.com (M.M.); mtakaso@kitasato-u.ac.jp (M.T.); 2Shonan University Medical Sciences Research Institute, Nishikubo 500, Chigasaki City 253-0083, Kanagawa, Japan

**Keywords:** osteoarthritis, infrapatellar fat pad, mast cell, diabetes

## Abstract

Diabetes mellitus (DM) has been suggested as a potential risk factor for knee osteoarthritis (KOA), and its underlying mechanisms remain unclear. The infrapatellar fat pad (IPFP) contributes to OA through inflammatory mediator secretion. Mast cells’ (MCs) role in diabetic IPFP pathology is unclear. In 156 KOA patients, hemoglobin A1c (HbA1c) was stratified (HbA1c ≥ 6.5, *n* = 28; HbA1c < 6.5, *n* = 128). MC markers (*TPSB2*, *CPA3*) in IPFP were studied. Propensity-matched cohorts (*n* = 27 each) addressed demographic differences. MC-rich fraction (MC-RF) and MC-poor fraction (MC-PF) were isolated, comparing MC markers and genes elevated in diabetic skin-derived MC (*PAXIP1*, *ARG1*, *HAS1*, *IL3RA*). *TPSB2* and *CPA3* expression were significantly higher in HbA1c ≥ 6.5 vs. <6.5, both before and after matching. MC-RF showed higher *TPSB2* and *CPA3* expression than MC-PF in both groups. In the HbA1c ≥ 6.5 group, *PAXIP1* and *ARG1* expression were significantly higher in the MC-RF than MC-PF. However, no statistical difference in the evaluated genes was detected between the High and Normal groups in the MC-RF. Elevated *TPSB2* and *CPA3* levels in the IPFP of high HbA1c patients likely reflect higher numbers of MCs in the IPFP, though no difference was found in MC-specific markers on a cell-to-cell basis, as shown in the MC-RF comparison. These findings deepen our understanding of the intricate interplay between diabetes and KOA, guiding targeted therapeutic interventions.

## 1. Introduction

The worldwide prevalence of osteoarthritis (OA) and diabetes mellitus (DM) has significantly increased, impacting approximately 16% and 11% of the overall population, respectively [1,2,3]. Some epidemiological investigations have suggested an association between DM, hyperglycemia, and OA [4,5,6,7]. Notably, the influence of DM extends beyond the mere existence of OA, adversely affecting pain levels and functional abilities in individuals with OA. Furthermore, research indicates that DM correlates with heightened pain intensity and reduced walking speed among patients with knee OA [8,9,10,11,12,13,14,15,16]. Despite these findings, the exact underlying mechanism linking DM to OA remains unclear.

Metabolic impairments linked to adipose tissue dysfunction are critical considerations. Within the knee joint, the infrapatellar fat pad (IPFP) accommodates adipose tissue, situated intracapsularly and extrasynovially in proximity to the synovium, cartilage, and bone [17,18]. Its position within the joint space enables the IPFP to impact the pathophysiology of OA by releasing inflammatory signals, including interleukin (IL)-6 and tumor necrosis factor-alpha (TNF-α) [19,20,21]. Previous findings from our group have highlighted the association between hypercholesterolemia and an inflammatory state in the IPFP [22]. However, the impact of diabetic conditions on IPFP pathology remains elusive.

Mast cells (MCs) have been identified as contributors to both chronic and acute inflammatory responses in synovium [23,24]. An increased presence of MCs within the synovium has been identified in individuals with knee OA (KOA), with a notable emphasis on those who are obese, suggesting their active role in inflammatory processes [25,26,27]. Although MCs normally exist in adipose tissue, there is an increased infiltration observed in abdominal visceral and subcutaneous adipose tissue in diabetic conditions [28,29]. This heightened presence contributes to the establishment of an inflammatory environment, as MCs possess the ability to release pro-inflammatory mediators [30,31,32]. However, it remains to be established whether there is an elevation in MC numbers in the IPFP of patients with diabetic KOA.

Previous epidemiological studies have consistently found a high prevalence of overweight and obesity in DM patients, and this may be the cause OA [33,34,35]. For instance, Fatani et al. reported that 41% of DM patients were overweight, and Hedley et al. found that 70% of DM patients were overweight or obese [34,35]. Furthermore, our earlier research indicated that 56% of OA patients with hemoglobin A1c (HbA1c) levels ≥ 6.5 fell into the obese category [36]. Therefore, connections between OA and DM were often attributed to the presence of obesity in individuals with DM [14]. In an effort to discern whether MCs contribute to the pathology of KOA in the context of DM, we compared MC marker expression in the IPFP between KOA patients with normal and high HbA1c concentrations using propensity-matched cohorts.

## 2. Results

### 2.1. Expression of TPSB2 and CPA3 between KOA Patients with HbA1c ≥ 6.5 and HbA1c < 6.5

IPFP specimens were harvested from 156 KOA patients. Table 1 presents a comparison of patient background factors in individuals within the High (HbA1c ≥ 6.5) and Normal (HbA1c < 6.5) groups who underwent quantitative polymerase chain reaction (qPCR) analysis before (Normal, *n* = 128; High, *n* = 28) and after propensity matching (Normal, *n* =27; High, *n* = 27). There were no significant differences in age, gender ratio, proportion of obesity, and Kellgren–Lawrence (KL) grade between the High and Normal groups (Table 1). Before propensity matching, the High group had significantly higher body mass index (BMI) and triglyceride (TG) levels and significantly lower total cholesterol (TCHO) levels compared to the Normal group (Table 1). Despite this, matched analysis did not reveal any differences between the two groups with High and those with Normal. The expression levels of *TPSB2* and *CPA3* in the High group were significantly higher than those in the Normal group, both before (*TPSB2*, *p* = 0.004; *CPA3*, *p* = 0.010) and after (*TPSB2 p* = 0.005, *CPA3*, *p* = 0.014, respectively; Figure 1A,B) propensity matching (Figure 1D).

### 2.2. Mast Cell Marker Expression in MC-Rich Fraction Derived from Normal and Diabetic Knee Osteoarthritis Patients

A total of 34 fresh IPFP samples obtained from KOA patients, comprising High (*n* = 12) and Normal (*n* = 34) groups, were utilized for magnetic isolation of MCs. Table 2 presents a comparison of patient background factors in individuals within the High and Normal groups who underwent magnetic isolation of MC before propensity matching (Normal, *n* = 22; High, *n* = 12) and after propensity matching (Normal, *n* = 8; High, *n* = 8). No significant differences were observed in age, gender ratio, proportion of obesity, and KL grade between the High and Normal groups, either before or after propensity matching (Table 2).

IPFP harvested from propensity-score-matched cohorts (High, *n* = 8; Normal, *n* = 8), were employed to assess and compare the gene expression among mast cells derived from patients with high and normal HbA1c levels. The analysis of MC-rich fraction (MC-RF) and MC-poor fraction (MC-PF) using magnetic beads showed higher expression levels of *TPSB2* and *CPA3* in the MC-RF than the MC-PF in both High (*TPSB2*, *p* < 0.001; *CPA3*, *p* < 0.001) and Normal groups (*TPSB2*, *p*= 0.003; *CPA3*, *p* = 0.009) (Figure 2A,B, respectively). However, in the High group but not the Normal group, *ARG1* and *PAXIP1* expression was significantly higher in MC-RF than MC-PF (*ARG1*, *p* = 0.021; *PAXIP1*, *p* = 0.009). There was no difference in *HAS1* and *IL3RA* expression in both High and Normal groups. However, no statistical difference in the evaluated genes was detected between the High and Normal groups in the MC-RF.

## 3. Discussion

Previous studies reported increased MCs in obesity, evidenced by elevated tryptase concentrations in the serum of obese individuals [37,38]. In KOA patients, the heightened presence of synovial MCs in the context of obesity has been documented. A critical nexus emerges when considering the potential transition from obesity to diabetes. Our study underscores the higher BMI values in the diabetic group compared to the HbA1c < 6.5 group, hinting at the intricate relationship between these two metabolic conditions. The propensity-matched analysis further nuances this connection, revealing that the diabetic group exhibits a higher expression of MC markers. Recent research has indicated that β-tryptase, encoded by *TPSB2*, stimulates *IL1B* expression in synovial macrophages and fibroblasts derived from KOA patients [25]. Additionally, β-tryptase is implicated in modulating joint lubrication in OA through the cleavage of lubricin [39]. Taken together with previous studies, our findings provide insights into the potential role of MCs in the complex interplay between diabetes and KOA, implicating MCs in the complex pathophysiology of diabetes and KOA.

Metabolic conditions induce alterations in MC phenotype and function [40,41,42,43]. In diabetic animals, there was an elevated presence of MCs in the kidneys, and these cells were identified to release diverse mediators, including chymase, tryptase, and cathepsin Gs [40]. In individuals with diabetes, MCs were found to secrete various cytokines such as IL-6 and IFN, along with chemokines like eotaxin, monocyte chemoattractant protein 1 (MCP-1), and RANTES [42]. The unique contribution of our research lies in the examination of gene expression in the MC-RF derived from the IPFP of KOA patients. The significant elevation in these MC markers in diabetic conditions emphasizes the potential role of MCs in the inflammatory milieu of the IPFP. Recent single-cell analyses have revealed altered gene expression profiles in MCs derived from individuals with DM, specifically the upregulation of *PAXIP1* and *ARG1* [41]. In the HbA1c ≥ 6.5 group, *PAXIP1* and *ARG1* expression were significantly higher in MC-RF than MC-PF. However, no difference was found in *PAXIP1* and *ARG1* on a cell-to-cell basis as shown in the MC-RF comparison. Therefore, *PAXIP1* and *ARG1* expression in IPFP-derived MCs may not have a clear connection to DM or KOA.

However, it is crucial to acknowledge the limitations of our study. The cross-sectional design limits our ability to establish causation, and the sample size may impact the generalizability of our findings. Future research should employ longitudinal approaches to unravel the dynamic interactions between MCs, diabetes, and KOA. Moreover, functional assessments are warranted to elucidate the direct consequences of increased MC markers on the progression of KOA in individuals with diabetes.

## 4. Materials and Methods

### 4.1. Patients

The study adhered to the principles of the Declaration of Helsinki and obtained approval from the Institutional Review Board of Kitasato University (protocol code: B19-259; Date of approval: 27 January 2020). Written informed consent was secured from all participants, explicitly outlining their agreement to participate in the study and the utilization of their IPFP following surgery. The investigation focused on IPFP samples obtained from patients diagnosed with KOA through radiography, who subsequently underwent total knee arthroplasty at our institution. The IPFP was isolated by separating the harvested fat pad from the surrounding fibrous tissue. All IPFP specimens were promptly frozen in liquid nitrogen at −196 °C and then stored at −80 °C before RNA extraction. Fresh IPFP samples before freezing were used for the magnetic isolation of MCs.

Patients with KOA were categorized according to their HbA1c concentration, with groups defined as HbA1c ≥ 6.5 (High) and HbA1c < 6.5 (Normal) [43]. Propensity score matching was performed by age, sex, body mass index, cholesterol, triglycerides, and Kellgren/Lawrence grades, which were variables expected to affect the gene expression in IPFP. We performed a 1:1 nearest matching with a caliper set of 0.1.

### 4.2. Isolation of MC Using Magnetic Beads

Recent single-cell analysis has unveiled an elevated expression of *ARG1*, *IL3RA*, *PAXIP1*, and *HAS1* in MC derived from the skin of individuals with diabetes [41]. To delineate the characteristics of MC derived from diabetic KOA patients, we utilized magnetic bead methods to isolate both the MC-RF and MC-PF. This approach involved comparing MC markers (*TPSB2*, *CPA3*) and genes associated with elevated expression in MCs derived from diabetic skin (*PAXIP1*, *ARG1*, *HAS1*, *IL3RA*). Patients’ clinical characteristics are summarized in Table 2 by group.

The isolation of MCs was conducted using magnetic methods as previously described [25]. In brief, cells were obtained from collagenase-digested synovial samples originating from both the High and Normal groups. Following centrifugation, cells underwent interaction with a biotin-labeled antibody cocktail (anti-CD3, CD14, CD19, CD90), all acquired from BioLegend (San Diego, CA, USA). Subsequent to the reaction with streptavidin-conjugated magnetic beads (BD™ IMag Streptavidin Particles Plus—DM, BD Biosciences, Tokyo, Japan), the MC-RF was isolated through negative selection, while positive fractions were isolated as MC-PF. Gene expression levels were assessed using a qPCR instrument (Bio-Rad CFX Connect, Bio-Rad, Hercules, CA, USA).

### 4.3. Quantitative Polymerase Chain Reaction (qPCR) Analysis

To explore whether there is an elevation in the expression levels of *TPSB2* and *CPA3* in KOA patients with high Hba1c values, we analyzed *TPSB2* and *CPA3* in the IPFP of KOA patients categorized into High and Normal groups. The procedures for total RNA extraction, cDNA synthesis, and qPCR utilizing SYBR Green are detailed in a previous study [27]. PCR primers for *TPSB2*, *CPA3*, and *GAPDH* were used for qPCR in our previous studies [27,44]. Primer sequences for *ARG1*, *IL3RA*, *PAXIP1*, and *HAS1* are shown in Table 3. We compared the expression of *TPSB2, CPA3*, and *GAPDH* in the IPFP between the two HbA1c groups. Relative expression was calculated using the mean of all control samples (samples from IPFP from Normal group or MC-PF derived from Normal group in vitro).

### 4.4. Statistical Analysis

Statistical analysis was conducted using the SPSS 25.0 statistical package. The Mann–Whitney U test was employed for the analysis of continuous variables, while Fisher’s exact test was utilized for categorical variables. Statistical significance was defined as *p* < 0.05.

## 5. Conclusions

In conclusion, our study sheds light on the intricate interplay between MCs and DM in the context of KOA. Through the integration of our findings with existing research, we enhance our understanding of the molecular mechanisms that underlie the pathological processes in the IPFP. This deeper comprehension creates opportunities for targeted interventions, potentially paving the way for novel therapeutic strategies aimed at addressing the specific challenges faced by individuals with diabetic KOA.

## Figures and Tables

**Figure 1 ijms-25-00877-f001:**
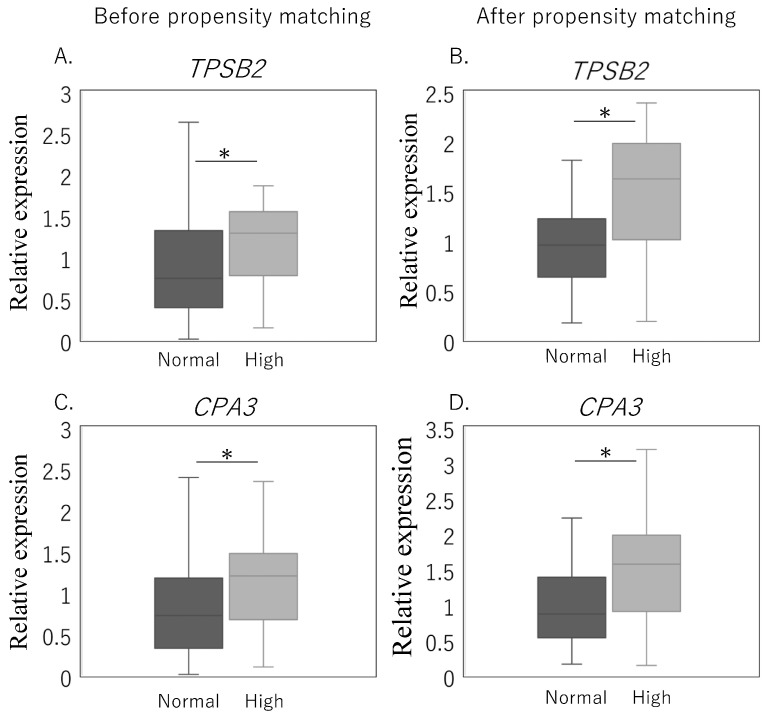
Effect of HbA1c value on *TPSB2* and *CPA3* expression in the infrapatellar fat pad before and after propensity matching. *TPSB2* (**A**,**B**), *CPA3* (**C**,**D**) expression in patients with High (HbA1c ≥ 6.5) and Normal (HbA1c < 6.5) before and after propensity matching. * *p* < 0.05 compared with the HbA1c < 6.5 group.

**Figure 2 ijms-25-00877-f002:**
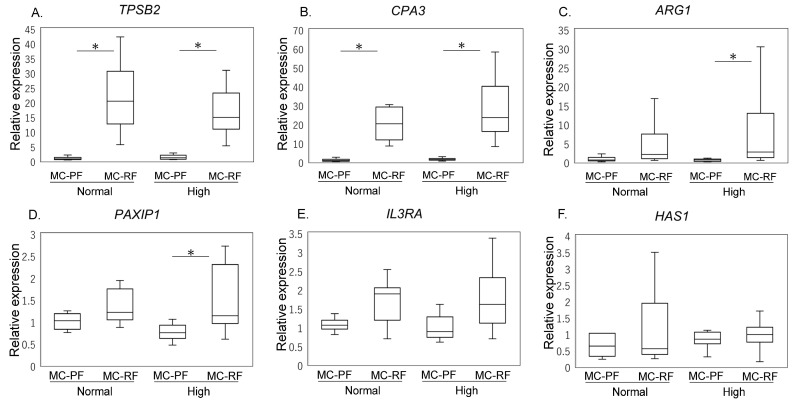
Characterization of mast cells derived from propensity-matched cohort. (**A**) *TPSB2*, (**B**) *CPA3*, (**C**) *ARG1*, (**D**) *PAXIP1*, (**E**) *IL3RA*, and (**F**) *HAS1* expression in the mast cell (MC)-poor fraction (MC-PF; THY-1+, CD3+, CD14+, CD19+) and MC-rich fractions (MC-RFs; THY-1-CD3-CD14-CD19-) derived from the infrapatellar fat pad of the High (HbA1c ≥ 6.5) and Normal (HbA1c < 6.5) groups. * *p* < 0.05.

**Table 1 ijms-25-00877-t001:** Patients’ clinical characteristics by HbA1c group used for qPCR analysis of infrapatellar fat pad.

	Before Match			After Match		
	Normal (N = 128)	High (N = 28)	*p* Values	Normal (N = 27)	High (N = 27)	*p* Values
Sex, female/male, N	97/31	21/7	0.93	19/8	20/7	0.761
Age, years	73.5 ± 8.5	71.7 ± 8.3	0.485	73.7 ± 7.9	71.9 ± 8.4	0.562
NW/OW/OB	52/55/21	7/11/10	0.053	7/15/5	7/10/10	0.264
BMI, kg/m^2^	26.3 ± 4.2	28.1 ± 4.0 *	0.028	27.8 ± 4.0	28.1 ± 4	0.869
TCHO (mg/dL)	209 ± 37	182 ± 34 *	<0.001	185 ± 41	184 ± 33	0.815
Triglyceride(mg/dL)	133 ± 84	179 ± 104	0.11	157 ± 90	180 ± 106	0.373
HbA1c	5.9 ± 0.3	7.0 ± 0.5 *	<0.01	5.9 ± 0.4	7.0 ± 0.5	<0.001 *
KL (2/3/4)	2/33/93	1/3/24	0.195	0/9/18	1/3/23	0.1

KL, Kellgren/Lawrence grade; BMI, body mass index; TCHO, total cholesterol. Data represent mean ± standard deviation or *n*. * *p* < 0.05 compared with the Normal (HbA1c < 6.5) group.

**Table 2 ijms-25-00877-t002:** Patients’ clinical characteristics by HbA1c group used for qPCR analysis of mast cells isolated from infrapatellar fat pad using magnetic beads.

	Before Match			After Match		
	Normal	High	*p*	Normal	High	*p*
	(N = 22)	(N = 12)	Values	(N = 8)	(N = 8)	Values
Sex, female/male, N	6/16	1/11	0.378	8/0	8/0	1.000
Age, years	72.2 ± 7.0	76.4 ± 5.3	0.080	74.0 ± 4.6	75.3 ± 5.4	0.626
NW/OW/OB	15/6/1	6/4/2	0.405	7/1/0	6/2/0	0.522
BMI, kg/m^2^	24.4 ± 2.7	26.3 ± 5.3	0.171	23.1 ± 1.9	24.0 ± 3.5	0.566
TCHO (mg/dL)	200 ± 32	191 ± 33	0.474	195 ± 32	202 ± 31	0.682
Triglyceride(mg/dL)	137 ± 62	150 ± 70	0.571	133 ± 57	155 ± 82	0.54
HbA1c	5.9 ± 0.3	6.8 ± 0.6 *	<0.001	5.8 ± 0.3	6.8 ± 0.2 *	<0.001
KL (2/3/4)	1/1/20	0/2/10	0.389	0/1/7	0/2/6	0.522

KL, Kellgren/Lawrence grade; BMI, body mass index; TCHO, total cholesterol. Data represent mean ± standard deviation or *n*. * *p* < 0.05 compared with the Normal (HbA1c < 6.5) group.

**Table 3 ijms-25-00877-t003:** Sequences of primers used in this study.

Primer	Sequence (5′-3′)	Product Size (bp)
*TPSB2*-F	CGCAAAATACCACCTTGGCG	138
*TPSB2*-R	GTGCCATTCACCTTGCACAC	
*CPA3*-F	GGCACTGACCTCAACAGGAA	71
*CPA3*-R	TCTGCACATGGGTCATTGGT	
*ARG1*-F	ACTCGAACAGTGAACACAGCA	71
*ARG1*-R	TTGTGATTACCCTCCCGAGC	
*IL3RA*-F	AGGCGTCAACAGTACGAGTG	157
*IL3RA*-R	CTGTGCAGGGGATACCGAAG	
*PAXIP1*-F	GGAGGTCAAGTATTACGCGGT	132
*PAXIP1*-R	TCTGGATTGTCCCCATCCTCT	
*HAS1*-F	TTGCAGCAGTTTCTTGAGGC	130
*HAS1*-R	GGGACCTGGAGGTGTACTTG	
*GAPDH*-F	TGCCACTCAGAAGACTGTGG	129
*GAPDH*-R	TTCAGCTCTGGGATGACCTT	

## Data Availability

The data presented in this study are available on request from the corresponding author.

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
