# Peer review of "Association between High HbA1c Levels and Mast Cell Phenotype in the Infrapatellar Fat Pad of Patients with Knee Osteoarthritis"

_ijms, 2024, doi:10.3390/ijms25020877_

Round 1

Reviewer 1 Report

Comments and Suggestions for Authors

This is straightforward manuscript attempting to determine whether changes in skin mast cell gene expression also occur in diabetic osteoarthritic knee infrapatellar fat pad mast cells. The premise being that abnormal gene expression associated with the mast cells may underlie osteoarthritis pathology. Experimental descriptions require clarification as detailed below. For instance, were all the synovium samples measured for gene expression? Were only the propensity score selected samples used for mast cell isolation and analysis? See more below.

There are however some very significant issues that need to be clarified.

1.      Patient populations: the criteria used to determine “propensity score” and how this was applied to segregate patient samples needs to be described.

2.      There are no male patient samples in the propensity score selected groups. This point should be stated and explain why no males were included.

3.      Clarification between when synovium vs infrapatellar fat pad was used should be clarified.

4.      When synovium was used, did the sample include infrapatellar fat pad or was the sample just the synovial membrane?

5.      How was the infrapatellar fat pad isolated for the other synovial tissues?

6.      Specimens were flash frozen in liquid nitrogen and stored at -80oC. So how could cells be isolated using magnetic cell separation methods? Ice crystal formation in the frozen cells would have destroyed the cytoplasmic membrane and thawing would have enabled RNase activity to destroy much of the RNA.

7.      The RTqPCR results report relative levels. However, given the concerns about isolating cells from frozen tissues for RNA isolation, it is very likely the mast cell-poor fraction did not contain or had very few intact cells. Thus the value of comparing the mast cell rich to mast cell poor fraction is not clear.

8.      The manuscript text needs to be thoroughly edited for clarity, grammar, and syntax.

9.      All abbreviations should be defined, such as TCHO.

10.  There is no Table 3 listing primer sequences.

Comments on the Quality of English Language

Multiple incomplete sentences were found, for instance, lines 17-19 in the abstract.

Poor syntax was noted in many places.

Missing details regarding methods and procedures significantly reduce manuscript clarity.

Author Response

  1. Patient populations: the criteria used to determine “propensity score” and how this was applied to segregate patient samples needs to be described.

Response:  Propensity score matching was performed by age, sex, body mass index, cholesterol, triglycerides, and Kellgren/Lawrence grades, which were variables expected to affect the gene expression in IPFP. We performed a 1:1 nearest matching with a caliper set of 0.1.(Line 183 - Line 186)

  1. There are no male patient samples in the propensity score selected groups. This point should be stated and explain why no males were included.

Response: A total of 34 fresh IPFP samples, comprising High (n = 12) and Normal (n=34) groups, were utilized for magnetic isolation of mast cells (MC). Male patients were included in the cohorts before propensity score matching; however, it is important to note that there are no male patient samples in the propensity score-matched groups. Additional information regarding the composition of the cohorts before propensity score matching has been incorporated into Table 2. (Line 105 – Line 111)

  1. Clarification between when synovium vs infrapatellar fat pad was used should be clarified.

Response: We analyzed IPFP. We apologize for the oversight. In the title of Figure 1, Table 1, and Table 2, the term "Synovium" should be read as "IPFP." Relative expression was calculated using the mean of all control samples (samples from “IPFP” from Normal group or MC-PF derived from Normal group in vitro). (Line 93, Line 98-99, Line 199)

  1. When synovium was used, did the sample include infrapatellar fat pad or was the sample just the synovial membrane?

Response: We apologize for the oversight. In the title of Figure 1, Table 1, and Table 2, the term "Synovium" should be read as "IPFP." We appreciate your attention to this clarification.

(Line 93, Line 98-99, Line 198)

  1. How was the infrapatellar fat pad isolated for the other synovial tissues?

Response: The infrapatellar fat pad (IPFP) was isolated by separating the harvested fat pad from the surrounding fibrous tissue. (Line 163 – 164)

  1. Specimens were flash frozen in liquid nitrogen and stored at -80oC. So how could cells be isolated using magnetic cell separation methods? Ice crystal formation in the frozen cells would have destroyed the cytoplasmic membrane and thawing would have enabled RNase activity to destroy much of the RNA.

Response: A total of 34 fresh IPFP samples, comprising High (n = 12) and Normal (n=34) groups, were utilized for magnetic isolation of MCs. (Line 105 – 106).

  1. The RTqPCR results report relative levels. However, given the concerns about isolating cells from frozen tissues for RNA isolation, it is very likely the mast cell-poor fraction did not contain or had very few intact cells. Thus the value of comparing the mast cell rich to mast cell poor fraction is not clear.

Response: A total of 34 fresh IPFP samples, comprising High (n = 12) and Normal (n=34) groups, were utilized for magnetic isolation of MCs. (Line 105 – 106).

  1. The manuscript text needs to be thoroughly edited for clarity, grammar, and syntax.

Response: We have corrected the syntax in various sections to enhance clarity.

  1. All abbreviations should be defined, such as TCHO.

Response: We have ensured that all abbreviations, including "TCHO," are now defined in the manuscript.

  1. There is no Table 3 listing primer sequences.

Response: We apologize for this oversight. Table 3, listing primer sequences, has been added to address this omission.

Comments on the Quality of English Language

Multiple incomplete sentences were found, for instance, lines 17-19 in the abstract.

Poor syntax was noted in many places.

Missing details regarding methods and procedures significantly reduce manuscript clarity.

Response: We appreciate the reviewer's feedback and have thoroughly revisited the manuscript to address the highlighted issues. We have corrected the syntax in various sections to enhance clarity. Additionally, we have provided missing details regarding methods and procedures to improve the overall quality of the manuscript.

Reviewer 2 Report

Comments and Suggestions for Authors

Dear Author
It was a pleasure to review this paper.
The paper addresses an actual topic, however shows several weaknesses which need to be rectified before considering it for publication. Please check all the acronyms all over the text.

Abstract:

"Diabetes mellitus (DM) is a recognized knee osteoarthritis (KOA) risk factor with unclear mechanisms." please re-write the sentence. In example "DM has been suggested as a potential..."

Introduction:

Line 31-31 "Numerous studies have established a connection between obesity-related overload conditions and an increased risk of developing knee osteoarthritis (KOA) [1-4]. Never-theless, when examining the epidemiological relationship between overweight or obesity and hand osteoarthritis (OA), it is conceivable that systemic factors implicated in the correlation with knee osteoarthritis (KOA) could play a role in the wider context of OA pathology [5-9]." This part is confusing since the topic of manuscript is not obesity

Line 43: osteoarthritis (OA): the acronym has already been used in line 34.

Line 53: abdominal visceral (VAT) : the acronym does not make sense

Line 57: infrapatellar fat pad (IPFP). the acronym has already been used

Line 58: "Previous epidemiological studies have consistently found a high prevalence of overweight and obesity in DM patients": this may be the cause of OA

MATERIALS AND METHODS SECTION IS IN THE WRONG PLACE (Please correct)

Line 165: the number of patients should be placed in the results section

Line 180: mast cells (MC) the acronym has already been used

Results: please start describing the number of patients collected

Line 85-87: "To explore whether there is an elevation in the expression levels of TPSB2 and CPA3 in KOA patients with high Hba1c values, we analyzed TPSB2 and CPA3 in the IPFP of KOA patients categorized into High and Normal groups." This part would better fit in the M&M section

Discussion

Line 137: "n individuals " a letter "I" is missing

Conclusions:

Why the study provides a "novel insights"?

The paper presents methodological gaps, and does not add strong concept to current literature. Moreover, it presents several redundant concepts. The Discussion section should always connect to the introduction and to the rationale of the study being well organized and neatly written. The correlation between obesity, diabetes and OA should be better investigate. Many confounfing factors are still present.

Major changes are needed to be considered suitable for publication.

Comments on the Quality of English Language

Minor editing of English language required

Author Response

Abstract:

"Diabetes mellitus (DM) is a recognized knee osteoarthritis (KOA) risk factor with unclear mechanisms." please re-write the sentence. In example "DM has been suggested as a potential..."

Response: We appreciate the reviewer's feedback. We have revised the sentence as follows:

Diabetes mellitus (DM)has been suggested as a potential risk factor for knee osteoarthritis (KOA), and its underlying mechanisms remain unclear.

Introduction:

Line 31-31 "Numerous studies have established a connection between obesity-related overload conditions and an increased risk of developing knee osteoarthritis (KOA) [1-4]. Nevertheless, when examining the epidemiological relationship between overweight or obesity and hand osteoarthritis (OA), it is conceivable that systemic factors implicated in the correlation with knee osteoarthritis (KOA) could play a role in the wider context of OA pathology [5-9]." This part is confusing since the topic of manuscript is not obesity.

Response: Thank you for bringing this to our attention. We have revised the introduction section to address the concerns and provide more clarity on the focus of the manuscript (Line 31-39).

Line 43: osteoarthritis (OA): the acronym has already been used in line 34.

Response: We have corrected it.

Line 53: abdominal visceral (VAT): the acronym does not make sense.

Response: We have deleted the acronym.

Line 57: infrapatellar fat pad (IPFP). the acronym has already been used.

Response: We have corrected it.

Line 58: "Previous epidemiological studies have consistently found a high prevalence of overweight and obesity in DM patients": this may be the cause of OA

Response Following the reviewer's suggestion, we have revised the sentence: Previous epidemiological studies have consistently found a high prevalence of overweight and obesity in DM patients, and this may be the cause of OA. (Line 60)

MATERIALS AND METHODS SECTION IS IN THE WRONG PLACE (Please correct)

Response: We appreciate your attention to detail regarding the placement of the Materials and Methods section. In our submission, we intentionally followed the format prescribed by the International Journal of Molecular Sciences, positioning the Materials and Methods section after the Discussion section and before the Conclusion section.

Line 165: the number of patients should be placed in the results section

Response: Number of patients have been placed in the results section.

Line 180: mast cells (MC) the acronym has already been used

Response: The number of patients has been appropriately included in the results section.

Results: please start describing the number of patients collected

Response: We have added the number of patients in results section.

Line 85-87: "To explore whether there is an elevation in the expression levels of TPSB2 and CPA3 in KOA patients with high Hba1c values, we analyzed TPSB2 and CPA3 in the IPFP of KOA patients categorized into High and Normal groups." This part would better fit in the M&M section.

Response: Following your suggestion, we have relocated the sentences from the Results section to the Materials and Methods section. (Line 191-199)

Discussion

Line 137: "n individuals " a letter "I" is missing

Response: We have corrected it. (Line 135)

Conclusions:

Why the study provides a "novel insights"?

Response: We have revised conclusions to avoid overreaching. (Line 209-210)

The paper presents methodological gaps and does not add strong concept to current literature. Moreover, it presents several redundant concepts. The Discussion section should always connect to the introduction and to the rationale of the study being well organized and neatly written. The correlation between obesity, diabetes and OA should be better investigate. Many confounding factors are still present.

Major changes are needed to be considered suitable for publication.

Response: We have carefully revised the manuscript based on reviewers’ comments and believe that it is now suitable for publication as a communication article.

Round 2

Reviewer 1 Report

Comments and Suggestions for Authors

The revised manuscript is significantly more clear. However the points below should be addressed.

L26: “MCs from high HbA1c patients expressed heightened PAXIP1 and ARG1, providing insight into KOA pathology in diabetes.” This statement indicates that PAXIP1 and ARG1 expression was greater in the MC-RF from patients with high HbA1c levels as compared to normal patients. However, the data in Figure 2 do not support this indication. The data found no statistical difference between the HbA1c high vs normal MC-RF data. Indeed the mean values appear to be very similar. That differences were identified between MC-RF and MC-PF should be expected and has no clear connection to diabetes or knee OA.

L154-158: These 2 sentences highlights the authors interpretation that expression of PAXIP1 and ARG1 are greater in the high HbA1c MC-RF, when as noted above no statistical difference was detected between the high HbA1c MC-RF and the normal HbA1c MC-RF. As such, the authors need to lessen their interpretation of these data. Alternatively, the authors could highlight that the elevated TSB2 and CPA1 levels in the fat pad of high HbA1c patients likely reflects higher numbers of MC cells in the fat pad, though no difference was found in MC specific markers on a cell to cell basis as shown in the MC-RF comparison.

The manuscript requires minor editing.

Comments on the Quality of English Language

Minor editing is still required.

Author Response

The revised manuscript is significantly more clear. However the points below should be addressed.

L26: “MCs from high HbA1c patients expressed heightened PAXIP1 and ARG1, providing insight into KOA pathology in diabetes.” This statement indicates that PAXIP1 and ARG1 expression was greater in the MC-RF from patients with high HbA1c levels as compared to normal patients. However, the data in Figure 2 do not support this indication. The data found no statistical difference between the HbA1c high vs normal MC-RF data. Indeed the mean values appear to be very similar. That differences were identified between MC-RF and MC-PF should be expected and has no clear connection to diabetes or knee OA.

Response: Thank you for your thoughtful insights. In light of the reviewer's comments, we have revised the sentences to more accurately reflect the data and avoid overreaching conclusions. (Line 24 – 28)

L154-158: These 2 sentences highlights the authors interpretation that expression of PAXIP1 and ARG1 are greater in the high HbA1c MC-RF, when as noted above no statistical difference was detected between the high HbA1c MC-RF and the normal HbA1c MC-RF. As such, the authors need to lessen their interpretation of these data. Alternatively, the authors could highlight that the elevated TSB2 and CPA1 levels in the fat pad of high HbA1c patients likely reflects higher numbers of MC cells in the fat pad, though no difference was found in MC specific markers on a cell to cell basis as shown in the MC-RF comparison.

Response: Thank you for your valuable feedback. In response to the suggestion, we have revised the discussion section to accurately reflect the findings and temper the interpretation of the data. (Line 156 -162)